# The Impact of the COVID-19 Pandemic on Cutaneous Drug Eruptions in a Swedish Health Region without Lockdown

**DOI:** 10.3390/microorganisms11081913

**Published:** 2023-07-27

**Authors:** Maria Pissa, Sandra Jerkovic Gulin

**Affiliations:** 1Department of Dermatology and Venereology, Department of Dermatology and Sexual Health, Södersjukhuset, Karolinska University Hospital, 118 83 Stockholm, Sweden; maria.pissa@regionstockholm.se; 2Department of Dermatology and Venereology, Ryhov County Hospital, 551 85 Jonkoping, Sweden; 3Division of Cell Biology, Department of Biomedical and Clinical Sciences, Linkoping University, 581 83 Linkoping, Sweden

**Keywords:** COVID-19, cutaneous drug eruptions, Stevens–Johnson syndrome, toxicoderma, toxic epidermal necrolysis

## Abstract

The incidence of severe cutaneous drug eruptions during the COVID-19 period in Sweden has not been studied previously. Our aim was to compare the incidence of these skin reactions in a Swedish health region during the COVID-19 pandemic period with that of the year after: we conducted a retrospective, observational cohort study using data from a national registry of patients diagnosed with cutaneous drug eruptions during the pandemic in Sweden. We included the number of patients diagnosed with severe cutaneous drug eruptions at the Department of Dermatology in the Jonkoping health region during the COVID-19 pandemic (1 April 2020 to 31 March 2021) and the reference period (1 April 2021 to 31 March 2022). We examined the monthly occurrences of cutaneous drug eruptions in three dermatology clinics within the Jonkoping health region. The frequency of these eruptions was determined for two distinct time periods: during the pandemic and post-pandemic. The study included 102 patients with cutaneous drug eruptions: 29 patients were diagnosed during the COVID-19 pandemic period and 73 were diagnosed during the reference period. The difference in the number of cutaneous drug eruptions cases (*p*-value = 0.0001, 95% CI 1.4995–3.5500, OR 2.3072) during the pandemic period compared to the post-pandemic period was significant. To our knowledge, the impact of the pandemic on cutaneous drug eruptions has not been investigated in EU countries. The increasing and differentiation of the number of diagnosed cutaneous drug eruptions cases after the pandemic could be explained by the removal of COVID-19 restrictions and the more frequent health-seeking behavior during the post-pandemic period.

## 1. Introduction

In March 2020, the global healthcare system was confronted with an unprecedented crisis when the World Health Organization (WHO) declared the COVID-19 pandemic [1]. This declaration marked the beginning of a tumultuous period characterized by the rapid spread of the novel coronavirus and the subsequent collapse of healthcare infrastructures across multiple nations. The impact on specialist care services was particularly pronounced, as healthcare systems struggled to cope with the overwhelming burden of COVID-19 cases.

Sweden, like many other countries, experienced the profound effects of the pandemic; however, its approach to managing the pandemic differed from that of many other nations. On 24 January 2020, the first confirmed case of SARS-CoV-2 was identified at Ryhov County Hospital in Jonkoping, Sweden. Subsequently, the virus gained a foothold in the country, leading health authorities to implement various measures to mitigate its spread. In mid-March 2020, recommendations were issued, urging individuals aged 70 and above, as well as those belonging to high-risk groups, to adopt additional precautions such as practicing social distancing and adhering to strict hygiene protocols.

While the impact of COVID-19 on the healthcare system and general population has been extensively studied worldwide, there remains a dearth of research regarding the occurrence of drug eruptions during the pandemic in Sweden. Drug eruptions refer to adverse skin reactions or allergies triggered by different medicines. Understanding the occurrence and characteristics of drug eruptions during the COVID-19 pandemic is crucial for healthcare professionals to provide appropriate medical care and mitigate potential risks associated with drug-related adverse events. A spectrum of cutaneous reactions after mRNA COVID-19 vaccines has been reported, although mostly minor and self-limited [2,3,4].

Therefore, it is imperative to conduct a study to assess the occurrence and nature of drug eruptions during the COVID-19 pandemic in Sweden. Such research would provide valuable insights into the interplay between drug usage, immune responses, and viral infections, ultimately contributing to the development of enhanced therapeutic strategies and patient care protocols during times of healthcare crises. By identifying potential risk factors and patterns associated with drug eruptions in this unique context, healthcare professionals can better allocate resources and devise preventive measures, ultimately ensuring the well-being and safety of patients during this challenging time [4].

Through our retrospective analysis, we sought to elucidate any discernible variations in the frequency of cutaneous drug eruptions (CDEs) within the specified time periods. This investigation holds significant implications for understanding the interplay between drug usage, viral outbreaks, and the subsequent occurrence of adverse skin reactions. By acquiring a comprehensive understanding of the prevalence and characteristics of CDEs in the context of the COVID-19 pandemic, healthcare professionals can devise tailored strategies to mitigate risks and optimize patient care during similar healthcare crises in the future. This research holds significant value in shedding light on the impact of the pandemic on drug-induced skin reactions.

The primary aim was to assess and compare the incidence of CDE within Jonkoping County, a prominent health region in Sweden, during two distinct time periods: the 12-month span encompassing the COVID-19 pandemic (1 April 2020, to 31 March 2021), and the subsequent 12-month period following the pandemic (reference period). Investigating the incidence of CDEs during the COVID-19 pandemic in Sweden fills an existing knowledge gap in the literature. To our knowledge, no previous research has explored the impact of the pandemic on CDE occurrence in European Union (EU) countries, making our study unique in its contribution to the field of dermatology and drug safety. This study sought to provide critical insights into the prevalence and trends of CDEs in the context of the COVID-19 pandemic. Understanding the impact of the pandemic on CDE occurrence is vital for healthcare professionals to effectively manage these serious skin reactions and enhance patient safety.

## 2. Materials and Methods

This study was conducted as a retrospective, observational cohort study, focusing on three major hospitals: Ryhov County Hospital, Highland Hospital of Nassjo, and Varnamo Hospital. Jonkoping County, situated in the southeastern health region of Sweden, caters to a population of approximately 360,000 residents.

To gather comprehensive data, we analyzed the records of patients who received diagnoses of several systemic drug reactions including toxicoderma, toxic epidermal necrolysis (TEN), and Stevens–Johnson syndrome (SJS) at the Department of Dermatology within the Jonkoping health region. By identifying and analyzing cases of CDE, including severe and potentially life-threatening conditions such as TEN and SJS, we aimed to gauge the impact of the COVID-19 pandemic on the occurrence of drug-induced skin reactions.

The scope of our study extended over two distinct time periods, namely the COVID-19 pandemic period (1 April 2020 to 31 March 2021) and the subsequent post-pandemic period (1 April 2021 to 31 March 2022). This research endeavor was categorized as a quality review and underwent a thorough approval process from the Operations Manager and Head of the Department of Dermatology at Ryhov County Hospital, located in the Jonkoping region of Sweden. The study was conducted in strict adherence to Section 31 of the Health and Medical Services Act, as outlined in the publication Lakartidningen 2015; 112: C9CL, ensuring compliance with the ethical and legal regulations governing healthcare research. To identify relevant cases for our analysis, we employed the ICD-10 codes (International Statistical Classification of Diseases, version 2019). All cases identified as L27.0 Generalized skin eruption due to drugs and medicaments/toxicoderma, L51.1 Bullous erythema multiforme/Stevens–Johnson syndrome (SJS), L51.2 Toxic epidermal necrolysis (TEN)/Syndrome Lyell during pandemic and post-pandemic period were included in the study. Cases coded with the following codes were excluded from the study: L27.1 Localized skin eruption due to drugs or medicaments, L56.0 Drug phototoxic response, L56.1 Drug photoallergic response, L51.8 Other erythema multiforme, L51.9 Erythema multiforme, unspecified.

By employing the standardized coding system, we aimed to ensure consistency and accuracy in case identification across the study. This approach allowed us the capture and analysis of a comprehensive dataset of patients who experienced cutaneous drug eruptions within the specified time periods. By utilizing this methodology, we sought to establish a comprehensive understanding of the occurrence and characteristics of cutaneous drug eruptions during the COVID-19 pandemic and the subsequent post-pandemic period.

In this study, we performed a comprehensive analysis to assess the monthly occurrences of different forms of CDE with a focus on toxicoderma, SJS, and TEN across the three dermatology clinics operating within the Jonkoping healthcare region. Our investigation aimed to examine the impact of the COVID-19 pandemic on the frequency of CDE cases by comparing the pandemic period (1 April 2020 to 31 March 2021) to the subsequent post-pandemic period (1 April 2021 to 31 March 2022).

## 3. Results

Our analysis yielded compelling results, revealing a notable disparity in the frequency of CDE cases between the two defined time periods, the pandemic period (1 April 2020 to 31 March 2021) and the subsequent post-pandemic period (1 April 2021 to 31 March 2022). We observed a substantial increase in the number of reported CDE cases during the post-pandemic period. During the pandemic period, a total of twenty-nine cases of CDE were documented across the three dermatology clinics. However, in the post-pandemic period, the number of reported cases surged significantly, reaching a total of seventy-three. The disparity in case numbers between the two periods was statistically significant (*p* = 0.0001, 95% CI 1.4995 to 3.5500, OR 2.3072). (Figure 1). To better understand the total number of consultations, in Figure 2, we provide data with a total number of patient consultations (digital referrals vs. physical visits) during the pandemic (1 April 2020 to 31 March 2021) and the after-pandemic periods (1 April 2021 to 31 March 2022). 

## 4. Discussion

Within the current scientific landscape, there is a notable dearth of comprehensive research that specifically investigates the impact of the COVID-19 pandemic on the occurrences of CDE worldwide. Despite the increasing recognition of the pandemic’s far-reaching effects, limited attention has been directed towards studying the association between the pandemic and the incidence of CDE cases worldwide. This research gap highlights a significant knowledge deficit that necessitates rigorous scientific inquiry to bridge the existing information void.

One intriguing observation that warrants investigation is the rise in the frequency of diagnosed CDE cases following the pandemic. Understanding the factors contributing to this trend poses a challenging task. One plausible contributing factor is the relaxation of COVID-19 restrictions that occurred as countries gradually emerged from the acute phase of the pandemic. The easing of lockdown measures, resulting in increased mobility and resumption of social interactions, may have inadvertently facilitated higher exposure to infective agents and, consequently, various drugs and medications. This increased exposure to potentially causative agents could have influenced the occurrence of CDE.

The relationship between the relaxation of COVID-19 restrictions and the subsequent rise in CDE cases is complex and probably multifactorial. The increased mobility and social interactions may have led to a greater utilization of medications, including prescription drugs, over-the-counter medications, and self-administered remedies. The diverse pharmacological profiles and potential side effects of these medications create an intricate web of interactions and reactions within the human body, including the skin. Consequently, the risk of developing CDE may have been elevated as a result of this increased medication usage and probably polypharmacy.

The study’s main weakness lies in the absence of additional denominator data, such as CDE per consultation, CDE per cases, or CDE per patient-day. Without these data, it is difficult to draw definitive conclusions regarding the impact of the pandemic on medical care utilization and the true prevalence of CDEs during that period. As restrictions eased and vaccination rates increased, there might have been a release of pent-up demand for medical care, leading to an uptick in CDE diagnoses after the pandemic. To strengthen the study’s findings and better understand the true effects of the pandemic on CDE prevalence, further research is needed, considering the factors mentioned above, and analyzing data from before, during and after the pandemic.

Another potential weakness of the study is the lack of analysis on sex, age, and ethnicity in relation to severe cutaneous drug eruptions during the COVID-19 pandemic. Understanding how these demographic factors may influence CDE occurrence could provide valuable insights into possible risk factors and tailored preventive measures. Future studies that consider these variables may further enhance the understanding of the impact of the pandemic on CDE prevalence.

Furthermore, the indirect consequences of the pandemic, such as psychological stressors and changes in healthcare-seeking behaviors, may also contribute to the observed rise in CDE cases. The psychological distress experienced during the pandemic period, including anxiety, fear, and social isolation, might have a profound impact on immune function and overall health. These psychosocial factors, in combination with the use of medications, may increase susceptibility to CDEs.

Is there a role of COVID-19 vaccination or COVID infection in the occurrence of CDE? Studies have reported a wide spectrum of cutaneous, mostly self-limited and minor, side effects [2,3]. Considering the impact of COVID-19 infection and vaccination on CDE prevalence is essential [4]. Some COVID-19 patients may require treatment with medications that have side effects, and the interplay between the infection and these drugs could lead to a higher incidence of serious CDEs. Additionally, the immune response triggered by COVID-19 could potentially influence the body’s reactions to other drugs taken simultaneously, further impacting the occurrence of CDEs. Many questions are to be answered. The potential role of COVID-19 vaccination in the occurrence of CDE during the period after the COVID-19 pandemic is an important consideration. Currently, there is only limited scientific data available regarding the specific association between COVID-19 infection, COVID-19 vaccination and CDE. To provide concrete evidence and validate these hypotheses, additional studies should compare the rates of serious CDEs among individuals with the COVID-19 infection, vaccinated individuals, and those who have neither experienced infection nor received the vaccine.

However, the observed rise in CDE cases during the post-pandemic period is a complex phenomenon with potential interplay between multiple factors. Additional research is needed to delve deeper into the underlying mechanisms and risk factors associated with this trend. Epidemiological studies, encompassing diverse populations and different geographical regions, are needed to investigate the temporal trends and geographical variations in CDE occurrences during post-pandemic period.

Furthermore, the post-pandemic period might have influenced individual attitudes regarding seeking medical assistance resulting in drug prescription. It is plausible that the individuals who might have deferred non-urgent medical visits during the height of the pandemic due to fear of exposure or overwhelmed healthcare systems sought medical attention more promptly in the post-pandemic period. This shift in healthcare-seeking behavior could have contributed to the observed divergence in the frequency of diagnosed CDE cases.

Moreover, investigating the specific impact of COVID-19-related factors, such as the use of medications for COVID-19 treatment or prevention, immune dysregulation associated with the viral infection, or psychological stressors during the pandemic, may provide valuable insights into the etiology and pathogenesis of CDE in the context of the pandemic [4]. This knowledge will not only facilitate the development of preventive measures, early detection strategies, and optimal management protocols for CDE cases, but also contribute to the broader understanding of the interplay between pandemics, drug reactions, and health in general.

Moreover, comprehensive pharmacovigilance systems and post-marketing surveillance efforts should be strengthened to monitor and document CDE cases effectively. By collecting standardized and detailed data on medication use, drug reactions, and outcomes, researchers can gain valuable insights into the causative agents, risk factors, and clinical manifestations of CDE in the post-pandemic era. This information can help to guide clinical decision-making to minimize the occurrence and severity of CDE cases.

## 5. Conclusions

The scarcity of comprehensive research investigating the impact of the COVID-19 pandemic on the occurrences of CDE worldwide highlights a significant knowledge gap in the scientific community. The observed rise in the frequency of diagnosed CDE cases during post-pandemic period is probably the complex interplay between the relaxation of COVID-19 restrictions, increased medication usage, psychosocial factors and different immune reactions. By addressing this research gap, we can advance our understanding of the relationship between pandemics and cutaneous drug reactions, and ultimately enhance patient care, medication safety, and health measures.

## Figures and Tables

**Figure 1 microorganisms-11-01913-f001:**
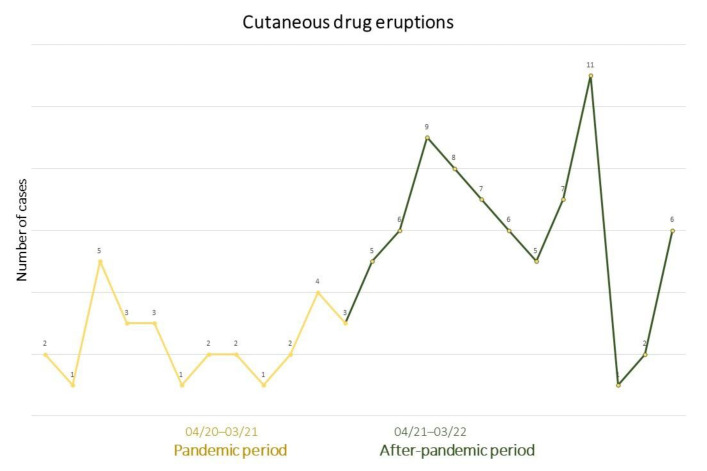
Number of cases of cutaneous drug eruptions during pandemic (1 April 2020 to 31 March 2021) and after-pandemic period (1 April 2021 to 31 March 2022).

**Figure 2 microorganisms-11-01913-f002:**
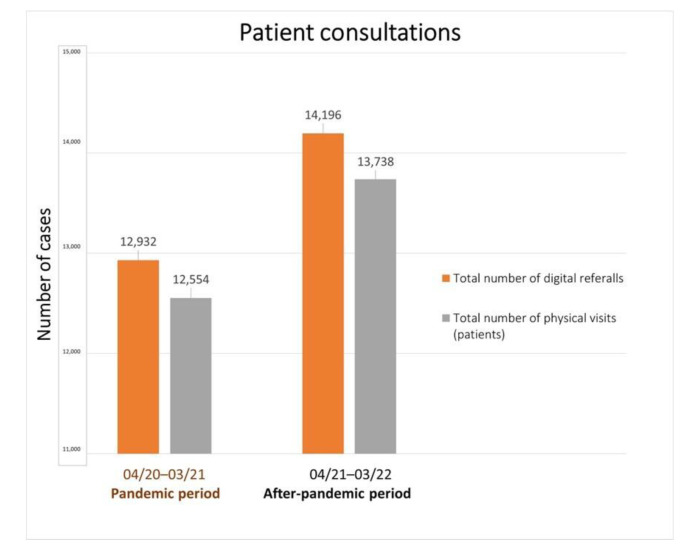
Total number of patient consultations (digital referrals vs. physical visits) during pandemic (1 April 2020 to 31 March 2021) and after-pandemic period (1 April 2021 to 31 March 2022).

## Data Availability

The data presented in this study are available on request from the corresponding author.

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
