# Peer review of "The Impact of the COVID-19 Pandemic on Cutaneous Drug Eruptions in a Swedish Health Region without Lockdown"

_microorganisms, 2023, doi:10.3390/microorganisms11081913_

Round 1

Reviewer 1 Report

The retrospective, observational cohort brief report titled "the impact of the COVID-19 pandemic on cutaneous drug eruptions in a Swedish health region without lockdown" conducted using data from a national registry of patients diagnosed with cutaneous drug eruptions during the pandemic in Sweden.  The report is short well written, and results were well presented but the paper lacks references in both introduction and discussion. 

Author Response

Thank you for your comments, we will provide references in both introduction and discussion part. 

Reviewer 2 Report

I congratulate the authors, the work is well-structured and scientifically valid. I find this very interesting given that there is an observed rise in the frequency of diagnosed CDE cases during the post-pandemic period. I just recommend adding the exclusion and inclusion criteria to the methods section and also adding to add a timeline image to show the entire process of research and the study outcomes.

Author Response

Thank you for your comments,

a)we have updated methods part with inclusion and exclusion criteria:

"All cases identified as L27.0 Generalized skin eruption due to drugs and medicaments/toxicoderma, L51.1 Bullous erythema multiforme/Steven-Johnson syndrome (SJS), L51.2 Toxic epidermal necrolysis (TEN)/Syndrome Lyell during pandemic and post-pandemic period were included in the study. Cases coded with following codes were excluded from the study: L27.1 Localised skin eruption due to drugs or medicaments were not included, L56.0 Drug phototoxic response, L56.1 Drug photoallergic response, L51.8 Other erythema multiforme, L51.9 Erythema multiforme, unspecified."

b) Figure 1 shows number of cases during different time periods (during april 2020 to march 2021, and during april 2021 to March 2022). 

Kind regrads

Reviewer 3 Report

this manuscript deals with the occurrence of  cutaneous drug
eruptions in three dermatology clinics within the Jonkoping health region during and after the corona pandemic and the associated lockdown phases or restrictions. The scientists' approach to comparing drug side effects with the reduced use of medical care caused by the pandemic is very interesting. The manuscript is well written and well developed, but improvement is still needed.

major points:

-The correlation of pandemic phases to the prevalence of CDEs is interesting, nevertheless further denominator data should be added. for example, CDE per consultation, CDE per cases, or CDE per patient-day. In the current context, it is only a supposition that the use of medical care was declining during the pandemic. Also a decrease in the absolute number of CDE diagnosed while the real numbers remained the same would also be possible. The addition of such reference data on medical care or, for example, mobility data from mobile phone use would significantly enhance the evaluation

minor comments:

page4: "Formulärets överkant
Formulärets nederkant"seems formatting leftover?

page 6: The statement "Conflicts of interest None disclosed" is misleading and should be reworded

Author Response

Thank you for your comments: 

-we will improve the manuscript

-denominator data that you suggest to add is impossible because we do not have access to the data on CDE per consultation, CDE per cases, or CDE per patient-day.  We have provided Figure 2 to visualize and compare total number of digital referrals and patient visit during and after pandemic 

Figure 2. Total number of patient consultations (digital referrals vs. physical visits) during pandemic (April 1, 2020, to March 31, 2021) and after-pandemic period (April 1, 2021, to March 31, 2022). 

This comment has also been added under discussion part as weakness of the study. 

-page4: "Formulärets överkant
Formulärets nederkant"seems formatting leftover?  This was not the authors misstake, it is corrected. 

-page 6: The statement "Conflicts of interest None disclosed" is misleading and should be reworded.   Not authors misstake, left maybe by accident. It is corrected. 

Reviewer 4 Report

I would like to congratulate the authors for the present report. Here goes a few comments:

The abstract does not require subheadings. Please check the authors guidelines.

The keywords are missing

The Introduction does not provide a clear rationale for the present study. Why is this study relevant and why should it be published?

The aim sentence (endo of the last Introduction paragraph) is not enough clear. What is the primary aim of the study?

Methods start with the aim sentence. This should be in the Introduction.

Can the authors provide more information regarding the demographics of the patients? Sex, age or major ethnicities?

How were the patients selected? All included? Random?

What was the sampling method?

May the authors debate the limitations, strength and validity of the study?

May the authors debate the generalization of the outcomes?

The reference list is not according to the journal guidelines.

Author Response

Thank you for the comments:

-the abstract does not require subheadings. - Done. Subheadings removed. 

The keywords are missing. - Done.  Keywords added. 

-The Introduction does not provide a clear rationale for the present study. Why is this study relevant and why should it be published? Done. We will add rationele.  "The primary aim was to assess and compare the incidence of CDE within Jonkoping County, a prominent health region in Sweden, during two distinct time periods: the 12-month span encompassing the COVID-19 pandemic (April 1, 2020, to March 31, 2021), and the subsequent 12-month period following the pandemic (reference period). Investigating the incidence of CDEs during the COVID-19 pandemic in Sweden fills an existing knowledge gap in the literature. To our knowledge, no previous research has explored the impact of the pandemic on CDE occurrence in European Union (EU) countries, making our study unique in its contribution to the field of dermatology and drug safety. This study sought to provide critical insights into the prevalence and trends of CDEs in the context of the COVID-19 pandemic. Understanding the impact of the pandemic on CDE occurrence is vital for healthcare professionals to effectively manage these serious skin reactions and enhance patient safety."

-The aim sentence (endo of the last Introduction paragraph) is not enough clear. What is the primary aim of the study? We will rewrite the sentence and made it clear. Done.

-Methods start with the aim sentence. This should be in the Introduction. Done

- Can the authors provide more information regarding the demographics of the patients? Sex, age or major ethnicities? Thank you for the comment but this cannot be provided for this paper but it will be included in our larger study with special ethical approval.  We will include this comment under discussion part as a potential weakness. " Another potential weakness of the study is the lack of analysis on sex, age, and ethnicity in relation to severe cutaneous drug eruptions during the COVID-19 pandemic. Understanding how these demographic factors may influence CDE occurrence could provide valuable insights into possible risk factors and tailored preventive measures. Future studies that consider these variables may further enhance the understanding of the impact of the pandemic on CDE prevalence. "

-How were the patients selected? All included? Random? What was the sampling method? - We will add inclusion and exclusion criteria in method part. This is restrospective observational study.  Done.

-May the authors debate the limitations, strength and validity of the study? - May the authors debate the generalization of the outcomes? We will include limitations of the study  and debate the generalisation of the outcome in the discussion part. Done. 

-The reference list is not according to the journal guidelines. -Changed. Done. 

Round 2

Reviewer 4 Report

Dear author, I have no more concerns.